# Fecal microbiota transplantation mitigates respiratory infection in rats exposed to hypobaric hypoxia by modulating the NLRP3 inflammasome and mucosal immunity

Huan Wang[1�], Zhenyan Wu[1�], Yaolei Zhang[2], Lijun Tang[3], Guangqing Shi[1], Lijie Ma[1*], Zhu Huang[4*], Jing Zhou[1*]

1 Department of Respiratory and Critical Care Medicine, General Hospital of Western Theater Command, Chengdu, Sichuan, P. R. China, 2 Basic Medical Laboratory, General Hospital of the Western Theater of Command, Chengdu, Sichuan, P. R. China, 3 Department of General Surgery, General Hospital of the Western Theater of Command, Chengdu, Sichuan, P. R. China, 4 Department of Hyperbaric Oxygen, General Hospital of the Western Theater of Command, Chengdu, Sichuan, P. R. China

☯ These authors contributed equally to this work.
* zhou8111027@foxmail.com (JZ); 13032877199@163.com (ZH); mljp321@163.com (LM)

## Abstract

### Objective

To investigate the role of the gut-lung axis in respiratory infection under hypobaric hypoxia and the therapeutic potential of fecal microbiota transplantation (FMT).

### Methods

Rats were exposed to hypobaric hypoxia (simulated 5000 m) for 14 days. Gut microbiota and serum short-chain fatty acids (SCFAs) were analyzed via 16S rRNA sequencing and GC-MS. Rats were then infected with *Streptococcus pneumoniae* and treated with FMT. Lung inflammation, NLRP3 inflammasome activity, cytokines, bacterial load, and secretory IgA (sIgA) were assessed.

### Results

Hypobaric hypoxia triggered gut dysbiosis, marked by reduced abundance of *Firmicutes* D and Lactobacillus, elevated *Bacteroidota*, and decreased SCFA levels.. FMT restored microbiota composition, increased acetic and butyric acid levels, and attenuated lung inflammation. FMT also enhanced NLRP3 inflammasome activation (NLRP3, ASC, Caspase-1), elevated IL-1β, IL-6, and TNF-α in BALF, reduced bacterial colonies, and increased airway sIgA in infected rats.

### Conclusions

FMT alleviates hypobaric hypoxia-aggravated respiratory infection by restoring gut microbiota, modulating SCFAs, and enhancing NLRP3-mediated mucosal immunity, highlighting the gut-lung axis as a therapeutic target.

**Data availability statement:** The datasets presented in this study can be found in online repositories. The names of the repository/repositories and accession number(s) can be found at: https://www.ncbi.nlm.nih.gov/, PRJNA1186348. All of the other data supporting the findings of this study are available from the corresponding author upon reasonable request.

**Funding:** This study was supported by the Chengdu Medical Research Project (Grant No.2024011). The funders had no role in study design, data collection and analysis, decision to publish, or.

**Competing interests:** The authors declare that they have no competing interests.

## 1. Introduction

The most significant change at moderate (> 1500 meters) to high altitude (> 2500 meters) is an exponential decrease in atmospheric pressure [1], which lowers the partial pressure of inspired oxygen ($PIO_2$), alveolar oxygen partial pressure ($PAO_2$), and arterial oxygen pressure ($PaO_2$)—a condition collectively termed hypobaric hypoxia. According to statistics, about 2% of the world's population lives at altitudes above 1,500 meters, where hypobaric hypoxia, cold and dry conditions, high levels of radiation, and the effects of climate change pose significant health challenges [2]. Acclimatization to altitude is a complex ability of organisms, primarily involving the cardiovascular and respiratory systems, to increase the supply of oxygen to body tissues [3]. High altitude environment present a risk of hypoxic exposure, and respiration plays an important role in compensating for hypobaric hypoxia at high altitude [4]. Studies have shown that at plateaus above 5,000 meters, individuals are unable to undergo long-term acclimatization as oxygen diffusion becomes a limiting factor to physical exercise capacity [3].This may be the cause of respiratory infections that are common at high altitudes, with common symptoms including sinusitis, pharyngitis, bronchitis and pneumonia [5].

The gut microbiota has become a research hotspot for revealing the mechanisms of disease onset and progression, which is closely related to immunity, hormonal and metabolic homeostasis, as well as to various diseases and specific environmental adaptations [6].The gut microbiota is susceptible to host and exogenous factors, in which exposure to hypobaric hypoxia in high plateau may lead to changes in the structure and function of the gut microbiota [7]. However, the results of previous studies on the effects of high-altitude environments on the gut microbiota are not entirely consistent. The study data support the altered gut microbiota in the group of hypoxic rats simulating acute high altitude exposure,as evidenced by significant increases in *Shannon*, *Simpson* and *Akkermansia*, as well as significant decreases in the *Firmicutes* to *Bacteroidetes* ratio and *Bifidobacterium* [8].It has also been shown that simulated plateau-exposed hypobaric hypoxia rats have an abnormal gut microbiota characterized by an increased abundance of *Parabacteroides*, *Alistipes*, and *Lactococcus genera*, as well as a greater ratio of *Bacteroides* to *Prevotella* ratio [9]. The reasons may be due to differences in altitude and temperature and humidity, as well as different types of experimental animals and different diets. Therefore, the changes in the gut microbiota under different conditions need to be further explored.

With the intensive study of the role of gut microbiota, the concept of the microbiome-gut-lung axis is emerging, providing new perspectives for research in the field of health and respiratory diseases [10].Many studies have shown that gut microbiota dysbiosis may contribute to the development of a wide range of respiratory diseases, as the gut microbiota has a major impact on the maturation of immune cells and resistance to pathogens [11].Studies have shown that airway epithelial barrier function is closely related to respiratory infections, and hypoxia in the respiratory system causes epithelial barrier dysfunction, leading to a decrease in the body's ability to resist various pathogenic bacteria [12].The novelty of recent years has been the association between NLRP3 inflammasome activation and the development of

inflammatory diseases of the airways [13].The NLRP3 inflammasome composed of the innate immune receptor protein NLRP3, adapter protein ASC, and inflammatory protease Caspase-1 can activate pyroptosis [14].Pyroptosis destroys pathogen-infected cells, promotes the release of IL-1β and IL-18, activates the inflammatory response, elicits the onset of an immune response against the pathogen in question, and in turn stimulates the body's immune system to clear the pathogen [15]. Additionally,secretory immunoglobulin A (sIgA), synthesized by plasma cells, is the body's first line of defense against infection and is able to fight bacteria, viruses, and neutralize toxins, thereby protecting the host from microbial pathogens [16].It is thus clear that the clearance of viruses and bacteria from the upper respiratory tract is inextricably linked to the moderate activation of NLRP3 inflammasome and the immune defense role of sIgA. However, the gut microbiota's effect on NLRP3 inflammasome activity and airway mucosal immune barrier function under respiratory infection induced by high altitude exposure is unclear.

In summary, the aim of this study was to investigate the effects of simulated plateau exposure on the gut microbiota and its metabolites SCFA in rats. Specifically, we sought to delineate the changes in microbial community structure under hypobaric hypoxia and the consequent alterations in SCFA profiles.At the same time, we hypothesized that plateau-induced gut microbiota dysbiosis would affect airway mucosal barrier function,potentially through the disruption of microbial metabolite signaling.Therefore, we observed the effects of supplemental gut microbiota on NLRP3 inflammasome activity and inflammatory factor levels during simulated plateau-induced airway bacterial infection to elucidate the mechanistic link within the gut-lung axis, examining how gut-derived signals modulate respiratory mucosal immunity during high altitude respiratory infection.

## 2. Materials and methods

### 2.1. Animal treatment and sample collection

36 Sprague-Dawley (SD) rats (180–200 g, GemPharmatech Co., Ltd) were randomly selected. After 7 days of acclimatization under an atmospheric pressure environment (500 m above sea level and 20–24°C), the rats were divided into six groups: normoxic normobaric (NN), hypobaric hypoxia (HH), normobaric normobaric + *Streptococcus pneumoniae infection* (NN + SPI), hypobaric hypoxia + *Streptococcus pneumoniae infection* (HH + SPI), hypobaric hypoxia + fecal microbiota transplantation (HH + FMT), hypobaric hypoxia + fecal microbiota transplantation + *Streptococcus pneumoniae infection* (HH + FMT + SPI). The low-pressure chamber was used to simulate a plateau environment at an altitude of 5,000 meters above sea level (10% $O_2$, 80% humidity and 20°C).Donor rats (from the NN group) were selected based on normal health status, consistent body weight, and absence of enteric pathogens. Fresh fecal samples from three donor rats were pooled, homogenized in sterile saline, centrifuged, and the supernatant was used for transplantation to ensure microbial diversity and consistency.The HH + FMT group was included to assess the specific effects of fecal microbiota transplantation on gut microbiota composition and systemic inflammation in hypobaric hypoxia conditions, independent of bacterial infection. This group serves as a control to distinguish between the effects of hypoxia alone and those mediated by microbiota restoration.The altitude of 5000 meters was selected to simulate extreme high-altitude conditions, which are known to induce significant hypoxic stress and systemic physiological alterations, thereby providing a robust model to study gut-lung axis interactions under severe hypoxia [17,18].

The HH, HH + SPI, HH + FMT, and HH + FMT + SPI groups were housed in low-pressure chamber for 14 days. The NN and NN + SPI groups were continued under an atmospheric pressure environment for 14 days. A 12h/12h light-dark cycle was provided during the experimental period, and rats had free access to food and water.

After 14 days, the HH + FMT and HH + FMT + SPI groups were gavaged for 3 days with 2 mL of enteric antimicrobials consisted of 100 mg/kg cefazolin sodium (054220351, CSPC ZHONGNUO PHARMACEUTICAL Co., Ltd., China), 50 mg/kg vancomycin hydrochloride (1010202002, VIANEX S.A., Greece), and 100 mg/kg metronidazole sodium chloride (C2211002, Sichuan Pacific Pharmaceutical Co., Ltd., China). Then, the feces of rats in the NN group were made into a 200 mg/mL suspension with 0.9% saline, centrifuged at 2000 g for 10 min, and the supernatant was taken.This fresh fecal

 

supernatant preparation method was adapted from established protocols for rodent FMT studies [19].A sterilized catheter was inserted into the anus of rats to 5 cm to inject 5 mL of supernatant, and the rats were kept upside down for 5 min. Continuous enemas were operated for 7 days to reestablish the gut microbiota. Meanwhile, the NN + BI group was given 0.9% saline [19]. This antibiotic pretreatment was performed to deplete the existing gut microbiota, thereby creating a niche for the subsequent engraftment of the donor microbiota via FMT and ensuring a more pronounced and assessable effect of the transplantation.

After a 7-day period following the completion of FMT to allow for gut microbiota stabilization,the nostrils of rats were disinfected with 75% ethanol, and the upper respiratory tract was intranasally administrated with 200 μL bacterial solution containing $10^4$ CFU/mL *Streptococcus pneumoniae* (TS342380, Testo Biologicals, China) to rats in the NN + SPI, HH + SPI, and HH + FMT + SPI groups. After 3 days of continuous bacterial infection, broncho-alveolar lavage fluid (BALF) was obtained by needle cricothyroidotomy in rats anesthetized with 0.6% sodium pentobarbital (100 mL/kg, B21882, Baoji Herbest Bio-Tech Co.,Ltd., China). Then blood was collected from the orbital venous plexus and centrifuged for serum. The cecal contents, airway and lung tissue of rats were removed immediately after euthanasia by cervical dislocation. Except for the lung tissue, which was fixed in 4% paraformaldehyde (24018549, Biosharp, China), the rest of the tissues were rapidly frozen in liquid nitrogen for 1 h and subsequently stored at −80 °C for measurement.

## 2.2. Ethics approval

All animal experiments were approved by the Ethics Committee of the General Hospital of the Western Theater Command (approval no. 2023EC5-ky050) and conducted in accordance with the ARRIVE guidelines and the National Research Council's Guide for the Care and Use of Laboratory Animals.

## 2.3. 16S rRNA gene sequencing and analysis

PCR products were analyzed by 2% agarose gel electrophoresis, and the target fragments (~468 bp corresponding to the V3-V4 region) were excised under UV illumination using a sterile blade. The excised bands were purified using the DNA Gel Recovery Kit (AP-GX-250, Axygen, USA) according to the manufacturer's instructions.Cecal contents DNA were extracted using the Omega M5635 soil DNA Kit (D5625-01, Omega Bio-Tek, Norcross, GA, USA). DNA quality and integrity were checked by 0.8% agarose gel electrophoresis and ultraviolet spectrophotometer (Invertigen, USA). A fragment of the highly variable V3-V4 region of the bacterial 16S rRNA gene with a length of about 468 bp was used for sequencing. The primers were respectively 338F (5'-barcode + ACTCCTACGGGGAGGCAGCA-3') and 806R (5'-GGACTACHVGGGTWTCTAAT-3'). PCR products were analyzed by 2% agarose gel electrophoresis, and the target fragments were cut and recovered by DNA Gel Recovery Kit (AP-GX-250, Axygen, USA). Subsequently, Quant-iT PicoGreen dsDNA Assay Kit (AN82752A, Takara Bio, Japan) was used for the quantification of the recovered products by microplate reader (FLx800; BioTek, Winooski, VT, USA). Library construction was performed using the TruSeq Nano DNA LT Library Prep Kit (FC-121–4001, Illumina, USA). Agilent High Sensitivity DNA Kit (Agilent, USA) was used for library quality control by Agilent 2100 Bioanalyzer (G2938A, Agilent, USA). Promega QuantiFluor (Promega, USA) and Quant-iT PicoGreen dsDNA Assay Kit were used for library quantification. Paired-end 2 × 250-bp sequencing was performed by using the NovaSeq 6000 SP Reagent Kit (500 cycles) (MS-102–3003, Illumina, USA) relying on the NovaSeq platform. Bioinformatics analysis was performed according to the official tutorials (https://docs.qiime 2.org/2019.4/tutorials/) adapted and refined for QIIME2.

## 2.4. Quantification analysis of gut microbial SCFA

An appropriate amount of serum sample were taken in a 2 mL EP tube. 50 μL of 15% phosphoric acid, 100 μL of internal standard solution (isocaproic acid: 125 μg/mL) and 400 μL of ether were added, and the samples were vortexed for 1 min. Then, the samples were centrifuged at 12000 rpm for 10 min at 4°C, and the supernatant was extracted for detection. All

samples were analyzed using a Thermo Trace 1300 gas chromatograph (Thermo Fisher Scientific, USA) tandem with a Thermo ISQ 7000 mass spectrometer (Thermo Fisher Scientific, USA) using an electron impact ionization source in selective ion monitoring mode, and the electron energy was 70 eV. The chromatographic column was an Agilent HP-INNOWAX capillary column (30 m × 0.25 mm ID × 0.25 μm) with a split injection (injection volume, 1 μL; split ratio, 10:1). Temperature-programmed process: 90°C, 10°C/min to 120°C, 5°C/min to 150°C, 25°C/min to 250°C (2 min). The carrier gas was helium with a flow rate of 1.0 mL/min.

## 2.5. NLRP3 inflammasome activity and inflammatory cytokine assay

Western blotting was used to detect the expression levels of NLRP3, ASC and Caspase-1 proteins in rat airway mucosa tissues. An appropriate amount of milled anterior airway mucosa tissues were taken to a 2 mL EP tube. After adding lysis solution (P0013B, Beyotime, China) to lysate the tissues for 30 min, centrifuge the tissues at 12000 rpm for 15 min, and then determine the total protein amount of the supernatant by BCA assay kit (240007003, Solarbio, China). Proteins were denatured and subjected to sodium dodecyl sulfate polyacrylamide gel electrophoresis (SDS-PAGE) for 2 h with a constant voltage of 120V for 80 min. A constant current of 300 mA for 80 min was used to transfer the protein blot to the PVDF membrane (R0SB83698, Millipore, USA). The blotting membrane was blocked with 5% BSA (330H053, Solarbio, China) for 2h at room temperature. The anti-NLRP3 primary antibody (080639650, Novus Biologicals,USA), anti-ASC primary antibody (B9304, Biodragon,China), anti-Caspase-1 primary antibody (1080367, Abcam,UK), and anti-GAPDH primary antibody (00148309, Proteintech,China) were incubated overnight at 4°C. Then the secondary antibody solutions (goat anti rabbit IgG-AC2405126662, goat anti mouse IgG-AC2401072385, Servicebio, China) were incubated for 2h at room temperature. Finally, electrochemiluminescence(ECL) solution (S2302301, Yeasen, China) was added to the blotting membrane and a gel imaging system (Azure300, Azure Biosystems, Germany) was used to develop the target proteins. ImageJ software (National Institutes of Health, Bethesda, Maryland) was used to analyze the grayscale values of each band, and GAPDH was used as control.

The expression levels of IL-1β (JL20884, Jianglai Biotechnology Co., Ltd., China), IL-6 (JL20896, Jianglai Biotechnology Co., Ltd., China) and TNF-α (JL13202, Jianglai Biotechnology Co., Ltd., China) were determined in the BALF of rats according to the manufacturer's instructions. The OD value of each sample was measured at 450 nm using a fully automatic microplate reader (Multiskan GO, Thermo, USA) and a standard curve was plotted to calculate the cytokine concentration.

## 2.6. Histopathologic analysis of the lung

Hematoxylin-eosin (HE) staining was performed as follows: after fixation with 4% paraformaldehyde for 48h, tissues were routinely dehydrated and embedded. Next, the tissues were sectioned at a thickness of 4 μm, baked, dewaxed and rehydrated, and immersed in hematoxylin for 3 min for staining. After washing in water, the sections were subjected to 1% hydrochloric acid-alcohol for 10s to remove the excess hematoxylin, and rinsed in running water for 10 min to return to blue. Then the sections were immersed in eosin staining solution for 5 s, rinsed to excess stain quickly under running water, dried naturally and sealed with neutral gum. Finally, lung tissue structure and cell morphology were observed under an ordinary light microscope (DM3000, Leica, Germany) to assess the staining effect.

## 2.7. Colony counting of streptococcus pneumoniae in BALF

The 10 μL of BALF was serially diluted and plated on Columbia blood agar supplemented with 5% sheep blood (a selective medium for *Streptococcus pneumoniae*), and incubated at 37°C under microaerophilic conditions (5% $CO_2$) for 48 h. Colonies exhibiting typical morphology (α-hemolysis, dome-shaped) were counted as *S. pneumoniae*.

To confirm the identity of colonies as *S. pneumoniae*, randomly selected colonies were subjected to Gram staining, catalase testing (negative), and optochin susceptibility testing. Only optochin-sensitive colonies were counted as *S. pneumoniae.*

## 2.8. Measurement of sIgA levels in rat airway

Enzyme-linked immunosorbent assay (ELISA) was performed to detect the sIgA levels of rat airway in the NN, HH, HH+SPI, NN+SPI, HH+FMT and HH+FMT+SPI groups. Rat airway tissues were liquid nitrogen quick-frozen and then milled well. An appropriate amount of ground tissues were taken in a 2 mL EP tube, 1 mL of phosphate solution (pH 7.4) was added, and the tissue was centrifuged at 3000 rpm for 20 min in a low-temperature centrifuge. Finally, the sIgA levels (H108-2–1, Nanjing Jiancheng Bioengineering Institute, China) in the rat airway were detected by using a fully automatic microplate reader according to the manufacturer's instructions.

## 2.9. Statistical analysis

Microbiome bioinformatic analysis was conducted using the QIIME [20] (version 1.7.0, Boulder, CO, USA)) and R package [21] (version 2.15.3, Auckland, New Zealand).Alpha diversity was used to evaluate the evenness and richness of bacteria taxonomic diversity by Wilcoxon rank-sum test, which was expressed by the Chao, Shannon and Observed_species index. Statistical analysis of the difference test was performed by one-way analysis of variance (ANOVA) using SPSS 23.0 software (IBM, New York, USA), and normality was identified by Kolmogorov-Smirnov test before ANOVA. $P < 0.05$ was considered to be of statistical significance.

# 3. Results

## 3.1. Hypobaric hypoxia influenced body weight, food intake,and hematocrit levels in Rats

After 14 days, rats under hypobaric hypoxia (HH, HH+SPI, HH+FMT, HH+FMT+SPI groups) showed significantly reduced food intake (e.g., HH: 9.4±0.8 g vs. NN: 17.2±1.8 g; $P < 0.05$) and weight loss (e.g., HH: −4.4±1.7 g vs. NN:+7.5±0.6 g; P<0.05), while their hematocrit levels were significantly elevated (e.g., HH: 64.8±4.6% vs. NN: 45.7±3.5%; $P < 0.05$) compared to normoxic normobaric (NN, NN+SPI) groups(Table 1).

## 3.2. Hypobaric hypoxia changed gut microbiota and its metabolite SCFA in rats

16S rRNA gene sequencing was used to compare the gut microbiota differences in rats after hypobaric hypoxia challenge. 15,718 operational taxonomic units (OTUs) were defined in 18 samples from NN, HH and HH+FMT groups. Top 10 bacteria in relative abundance of species at the phylum level belonged to *Firmicutes A*, *Firmicutes D*, *Bacteroidota*, and *Actinobacteriota*, with *Bacteroidota* and *Actinobacteriota* showing the most remarkable differences (Fig 1A). At the genus level, the most remarkable differences in the frequency distribution of species among groups were found in the *Ligilactobacillus*, *Lactobacillus*, and *Limosilactobacillus* (Fig 1C). The results of alpha diversity,which reflects both the richness (Chao, Observed_species) and evenness (Shannon) of microbial communities,showed that the Chao, Observed_species, and Shannon indices were significantly increased ($P < 0.05$) in the NN and HH+FMT groups than those in the HH group (Fig 1B). This findings suggested that FMT somewhat reversed the effects of hypobaric hypoxia challenge on the abundance and diversity of rat gut microbiota. Furthermore, the hypobaric hypoxia challenge reduced the levels of gut

**Table 1. Effects of hypobaric hypoxia on body weight, food intake and hematocrit level in rats.**

| Group | NN | HH | NN+SPI | HH+SPI | HH+FMT | HH+FMT+SPI |
|---|---|---|---|---|---|---|
| Food intake (g) | 17.2±1.8 | 9.4±0.8* | 15.2±1.5 | 9.3±1.1* | 8.9±0.5* | 8.7±0.6* |
| Hematocrit levels (%) | 45.7±3.5 | 64.8±4.6* | 44.9±4.2 | 65.3±5.6* | 67.3±6.7* | 69.3±6.1* |
| Body weight (g) | 7.5±0.6 | −4.4±1.7* | 5.9±1.1 | −5.3±0.6* | −4.5±2.8* | −7.3±2.7* |

NN: normobaric normoxia; HH: hypobaric hypoxia; NN+SPI: normobaric normoxia+*streptococcus pneumoniae* infection; HH+SPI: hypobaric hypoxia+*streptococcus pneumoniae* infection; HH+FMT: hypobaric hypoxia+fecal microbiota transplantation; HH+FMT+SPI: hypobaric hypoxia +fecal microbiota transplantation+*streptococcus pneumoniae* infection; *$P < 0.05$ represents statistically significant.

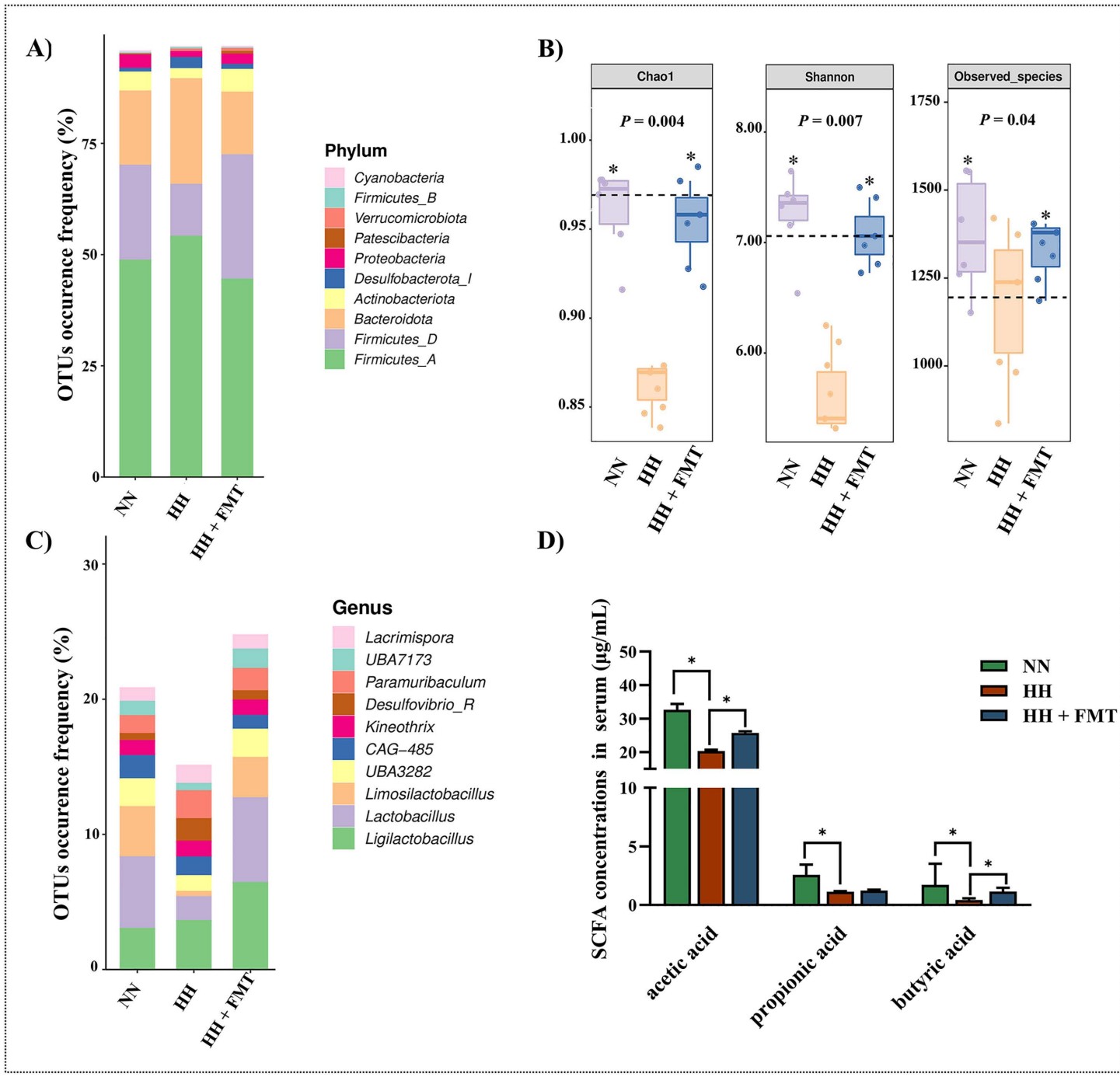

**Fig 1. FMT influenced the gut microbial community in rats after the hypobaric hypoxia challenge. A)** and **C)** Effects of FMT on taxonomic composition of the top 10 bacteria at the phylum and genus level; **B)** Effects of FMT on Alpha diversity of gut microbiota in rats; **D)** Effects of FMT on concentration of gut microbiota SCFA in rat serum. FMT: fecal microbiota transplantation; OTUs: operational taxonomic units; SCFA: short-chain fatty acids; NN: normobaric normoxia; HH: hypobaric hypoxia; HH+FMT: hypobaric hypoxia+fecal microbiota transplantation. *$P < 0.05$ represents statistically significant.

microbial metabolites such as acetic acid, propionic acid and butyric acid in the serum of rats (Fig 1D). FMT restored decreased concentrations of acetic acid and butyric acid ($P<0.05$), but no significant difference was observed in propionic acid concentrations (Fig 1D).These compositional shifts in the gut microbiota, characterized by a reduction in SCFA-producing genera (e.g., *Lactobacillus*) and an increase in opportunistic pathobionts, alongside the decline in systemic SCFA levels, suggest a compromised gut metabolic function and a potential disruption of gut-lung immune crosstalk under hypobaric hypoxia.

### 3.3. FMT alleviated lung inflammation in streptococcus pneumoniae-infected rats after hypobaric hypoxia challenge

HE staining of lung tissue showed that more inflammatory cells infiltrated around the airway and blood vessels of rats in the NN+SPI group after *streptococcus pneumoniae* infection compared with rats in the NN and HH groups. However, the degree of inflammatory cell infiltration in the HH+SPI group was more remarkable than in the NN+SPI group. The degree of inflammatory cell infiltration of lung in the HH+FMT+SPI group after gut microbiota supplementation was reduced compared with that in the HH+SPI group (Fig 2).

### 3.4. FMT restored NLRP3 inflammasome activity and inflammatory cytokine levels in streptococcus pneumoniae-infected rats after hypobaric hypoxia challenge

Compared with the NN group, protein expression of NLRP3, ASC, and Caspase-1 in airway epithelial tissue was elevated in the HH+SPI group following *Streptococcus pneumoniae* infection (P<0.05). However, these levels remained significantly lower than those in the NN+SPI group (P<0.05) (Fig 3A, B). After FMT, the expression of NLRP3, ASC, and Caspase-1 in the HH+FMT+SPI group was restored to levels comparable to the NN+SPI group (P<0.05)(Fig 3A; Fig 3B).

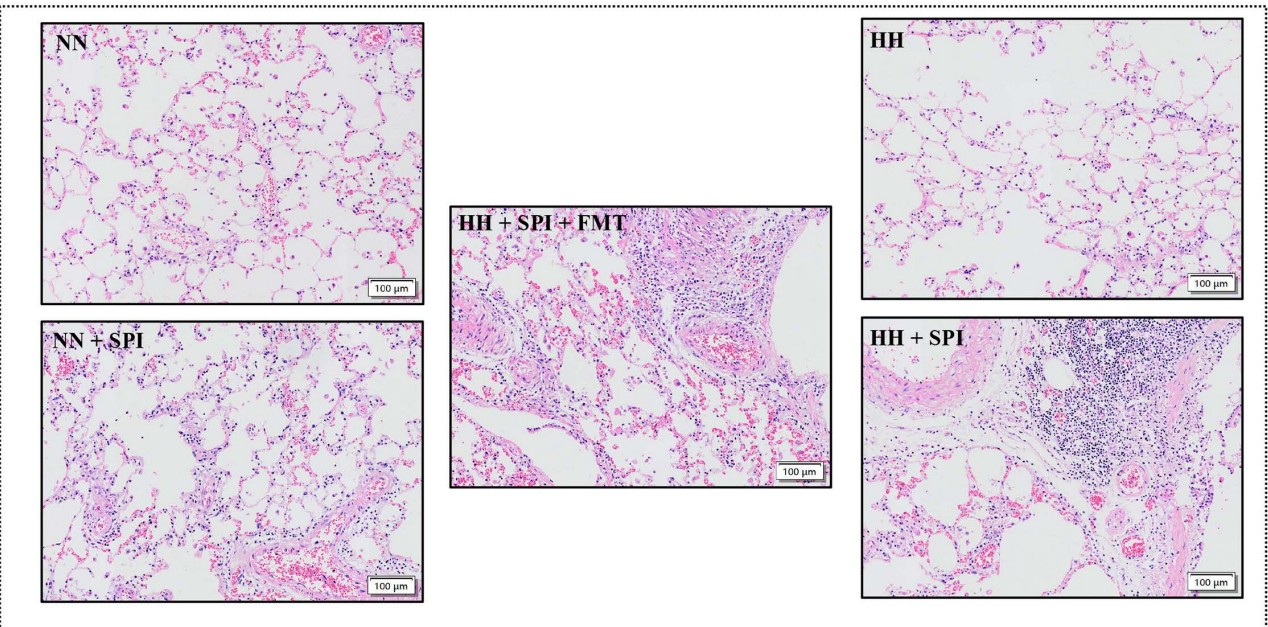

**Fig 2. FMT alleviated lung inflammation in *streptococcus pneumoniae*-infected rats after hypobaric hypoxia challenge.** NN: normobaric normoxia; HH: hypobaric hypoxia; NN+SPI: normobaric normoxia+*streptococcus pneumoniae* infection; HH+SPI: hypobaric hypoxia+*streptococcus pneumoniae* infection; HH+FMT+SPI: hypobaric hypoxia +fecal microbiota transplantation+*streptococcus pneumoniae* infection.

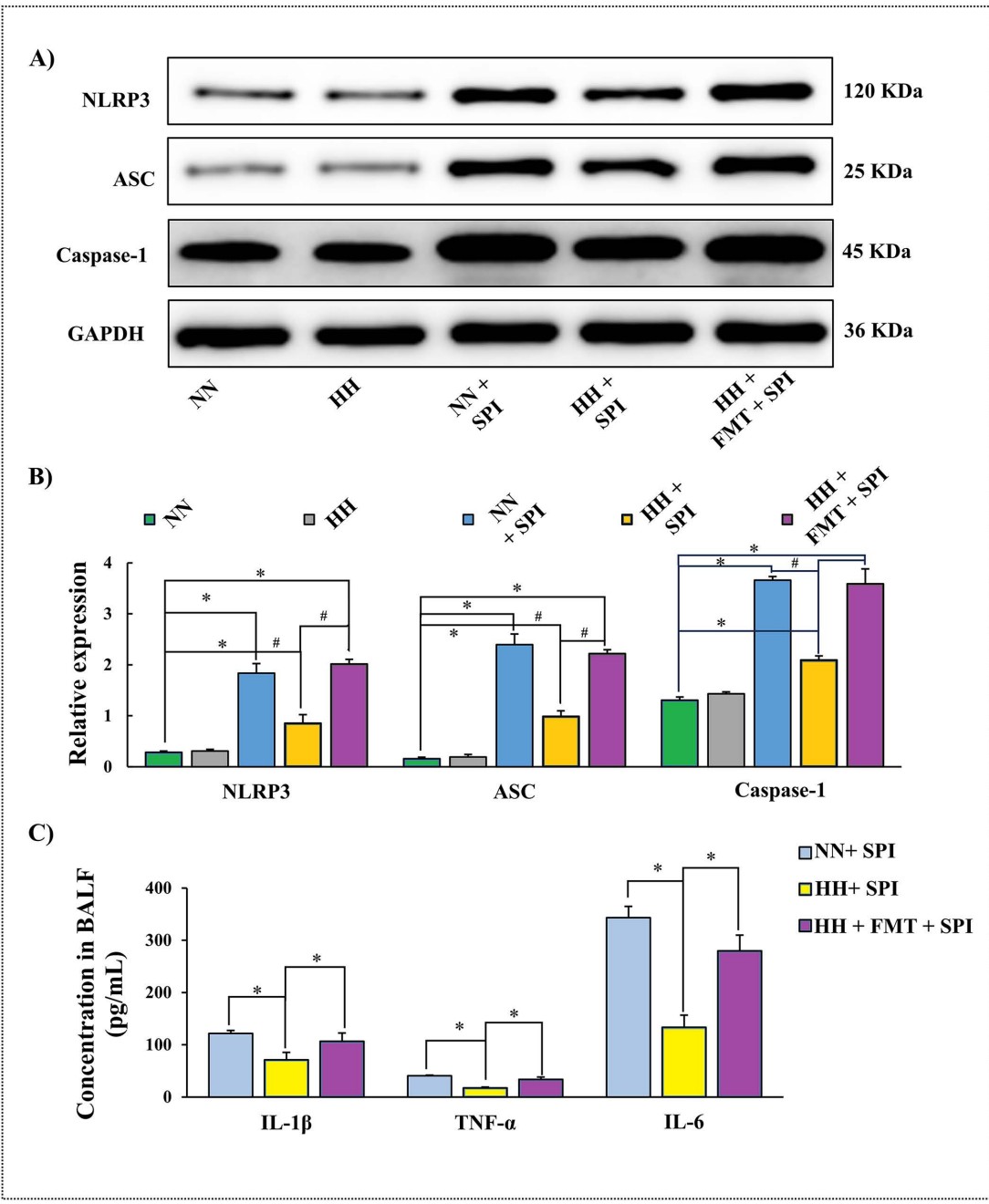

**Fig 3. FMT restored NLRP3 inflammasome activity and inflammatory cytokine levels in *streptococcus pneumoniae*-infected rats after hypobaric hypoxia challenge.** A) and **B)** Effects of FMT on NLRP3, ASC and Caspase-1 protein expressions in the airway mucosal tissues of rats; **C)** Effects of FMT on IL-1β, TNF-α, and IL-6 expression levels in BALF of rats. BALF: broncho-alveolar lavage fluid; NN: normobaric normoxia; HH: hypobaric hypoxia; NN+SPI: normobaric normoxia+*streptococcus pneumoniae* infection; HH+SPI: hypobaric hypoxia+*streptococcus pneumoniae* infection; HH+FMT+SPI: hypobaric hypoxia +fecal microbiota transplantation+*streptococcus pneumoniae* infection.

Similarly, levels of IL-1β, TNF-α, and IL-6 in BALF were lower in the HH+SPI group than in the NN+SPI group ($P<0.05$). FMT intervention (HH+FMT+SPI group) restored these cytokine levels to those observed in the NN+SPI group ($P<0.05$) (Fig 3C).

### 3.5. FMT reduced number of bacterial colonies in streptococcus pneumoniae-infected rats after hypobaric hypoxia challenge

After *streptococcus pneumoniae* infection, the number of bacterial colonies in the HH+FMT group under hypobaric hypoxia environment were significantly higher than that in the NN+SPI group under normobaric normoxia environment ($P<0.05$). The number of *streptococcus pneumoniae* colonies of rats in the HH+FMT+SPI group were more declined after FMT, but did not reach a significant difference compared with that in the HH+SPI group. (Fig 4).

### 3.6. FMT increased sIgA levels in streptococcus pneumoniae-infected rats after hypobaric hypoxia challenge

The sIgA levels in rat airway in the NN+SPI, HH+SPI and HH+FMT+SPI groups after *streptococcus pneumoniae* infection were higher than that in the NN group ($P<0.05$). However, under the same condition, the sIgA levels in the rat airway under hypobaric hypoxia environment (HH+SPI group) was lower than that in rats under normobaric normoxia environment (NN+SPI group) ($P<0.05$). FMT treatment (HH+FMT+SPI group) significantly reduced the colony count compared to the infected hypoxic controls (HH+SPI group) ($P<0.05$) (Fig 5).

## 4. Discussion

Previous studies have shown that the respiratory system is susceptible to pathogen infection when exposed to hypobaric hypoxia environment at high altitude, and that high altitude also affects the composition and diversity of the body's gut microbiota [22–24]. Unfortunately, the role of gut microbiota in plateau respiratory infection has been rarely reported to date. To address this issue, this study investigated the effect of gut microbiota on inflammatory response and airway mucosal immune function induced by airway bacterial infection at high altitude based on the microbiome-gut-lung axis.

First, the results of this study found that abnormal gut microbiota characterized by decreased abundance of *Firmicutes D* (phylum), *Actinobacteriota* (phylum), *Lactobacillus* (genus), and *Limosilactobacillus* (genus) and increased abundance

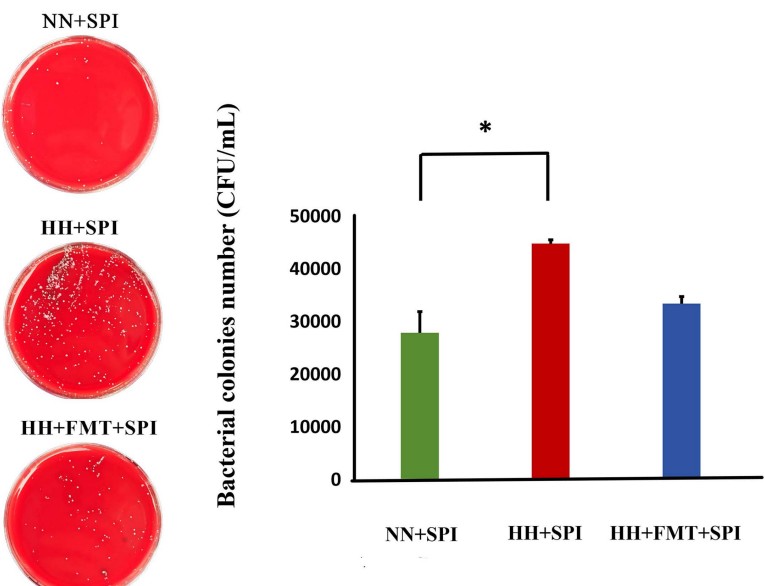

**Fig 4. FMT reduced number of bacterial colonies in *streptococcus pneumoniae*-infected rats after hypobaric hypoxia challenge.** NN+SPI: normobaric normoxia+*streptococcus pneumoniae* infection;HH+SPI: hypobaric hypoxia+*streptococcus pneumoniae* infection; HH+FMT+SPI: hypobaric hypoxia +fecal microbiota transplantation+*streptococcus pneumoniae* infection.

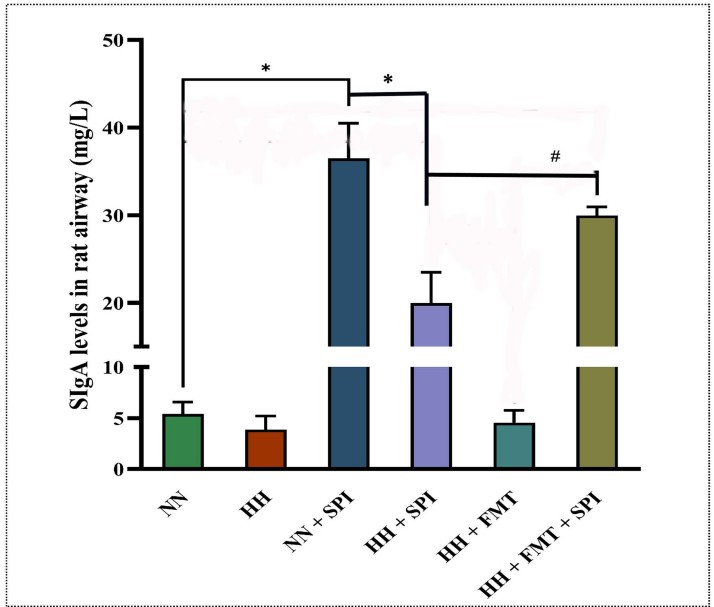

**Fig 5. FMT increased sIgA levels in *streptococcus pneumoniae*-infected rats after hypobaric hypoxia challenge.** sIgA: secretory immunoglobulin A; NN: normobaric normoxia; HH: hypobaric hypoxia; NN + SPI: normobaric normoxia + *streptococcus pneumoniae* infection; HH + SPI: hypobaric hypoxia + *streptococcus pneumoniae* infection; HH + FMT: hypobaric hypoxia + fecal microbiota transplantation; HH + FMT + SPI: hypobaric hypoxia + fecal microbiota transplantation + *streptococcus pneumoniae* infection.

of *Firmicutes A* (phylum), *Bacteroidota* (phylum) in hypobaric hypoxia rats. By FMT, the above abundance changes in gut microbiota were restored in our study. It has been demonstrated that the core microbiota altered in fecal samples from SD rats after 7 days of exposure to hypoxic condition at mid-altitude and plateau were identified as the phylum *Firmicutes*, genus *Akkermansia*, and genus *Lactobacillus* [25].Consistent with the results of the present study, some of high-throughput sequencing analyses also demonstrated significant changes in *Lactobacillus* and *Firmicutes* in the gut microbiota of rats exposed to plateau hypoxia [23,26,27].Microorganisms of the genus *Lactobacillus* are the most widely used probiotic, and are gram-positive, facultative anaerobic bacterium [28,29].There is evidence that administration of the probiotic *Lactobacillus* rhammosus GG improves respiratory asthma symptoms in mice and children [30,31]. Moreover, the main dominant bacterial phylum of the human gut microbiota is known to be *Firmicutes* [32].The effect of *Firmicutes* on respiratory diseases has not been reported in studies, but *Firmicutes* still has shown potential therapeutic and predictive value in other diseases such as inflammatory bowel disease, obesity and depressive disorder [33–35]. The above evidence suggests an important role for phylum *Firmicutes* and genus *Lactobacillus* in gut microbiota dysbiosis. However,the present study did not elucidate the exact mechanism by which *Lactobacillus* or *Firmicutes* mediates the gut-lung axis to influence respiratory immunity. This is a question that deserves further investigation.

Degradation of dietary fiber by intestinal microorganisms produces organic acids, gases, and large amounts of SCFA, with acetic acid, propionic acid and butyric acid being the main SCFA produced [36].SCFA play an important role in maintaining intestinal health by regulating intestinal luminal pH, mucus production, providing fuel for epithelial cells, and influencing mucosal immune function [37]. In the present study, we found that FMT was able to restore serum acetic acid and butyric acid levels in hypobaric hypoxia rats, suggesting that the replenishment of normal gut microbiota is essential for microbial SCFA production.Studies have shown that intestinal supplementation with many *Lactobacillus* species can increase SCFA levels in the intestine and blood [38,39].SCFA can reduce excessive inflammation and immunity in gastrointestinal and airway diseases by inhibiting histone deacetylase (HDAC) to increase the number and function of regulatory

T-cells (Treg), helper T-cells (Th)1, and Th17 effector cells [40].Moreover, SCFAs like butyrate act as signaling molecules through G-protein-coupled receptors (GPCRs, e.g., GPR41, GPR43) expressed on immune cells and lung epithelium, modulating the activation of the NLRP3 inflammasome and the production of cytokines and secretory IgA. In particular, the butyric acid has excellent anti-inflammatory properties and may help to alleviate the development of asthma and chronic obstructive pulmonary disease [41]. Butyric acid also induces apoptosis and inhibits proliferation of lung cancer cells in vitr [42,43]. Thus, butyric acid appears to be one of the mediators between the gut microbiota and the respiratory system, and our data support its role in the gut-lung axis, where its depletion under hypoxia may contribute to blunted NLRP3 activation and impaired mucosal defense.

Inflammasome induces an inflammatory response through Caspase-1 activation and protects the host from microbial pathogens [44]. To characterize the role of NLRP3 inflammasome activation in the inflammatory immune response, we observed the effects of timely supplementation of gut microbiota on the levels of NLRP3 inflammasome and inflammatory factors in hypobaric hypoxia rats during airway bacterial infection. As previously described, after *streptococcus pneumoniae* infection, hypobaric hypoxia rats with supplementation of gut microbiota (FMT) had activated expression of airway inflammasome proteins (NLRP3, ASC, and Caspase-1), and the levels of IL-1β, IL-6, and TNF-α in BALF were regulated. It has been reported that NLRP3 protects the pulmonary epithelial barrier during *Streptococcus pneumoniae* infection by increasing the adhesion of alveolar epithelial cells [45].Studies have shown that NLRP3 inflammasome activation plays an important role in regulating the host immune response during *Streptococcus pneumoniae* infection, and the mechanism involves the participation of the ALK/JNK/NEK7-NLRP3 signaling pathway [46]. Therefore, we hypothesized that the improvement of host immune function by FMT during airway bacterial infection in the plateau might be related to the activation of NLRP3 inflammasome. On the other hand,IL-1 cytokines (especially IL-1β and its receptor) play a protective role in murine pneumococcal pneumonia, protecting the host from *Streptococcus pneumoniae* infection [47].In studies targeting IL-6 and *Streptococcus pneumoniae* infection, elevated levels of IL-6 downregulate activation of the cytokine network to protect the host against pneumococcal pneumonia [48].sIgA is the main component of local mucosal immunity and the first line of defense to prevent pathogenic bacteria from penetrating the mucosa.The more sIgA on the mucosal surface, the greater the ability to neutralize viruses and fight infection [49]. Our results also revealed elevated levels of sIgA in hypobaric hypoxia and airway-infected rats after FMT, suggesting a role for gut microbiota in the maintenance of normal mucosal immune function in plateau respiratory infection.

Our findings that FMT simultaneously alleviated lung inflammatory cell infiltration yet enhanced NLRP3 inflammasome activity and proinflammatory cytokine levels may appear paradoxical.. However, this outcome likely reflects the restoration of a balanced and protective immune response in the hypoxic host. Hypobaric hypoxia blunts early innate immune recognition and signaling (manifesting as low NLRP3 activity and cytokine production),thereby permitting uncontrolled bacterial proliferation and subsequent dysregulated, compensatory tissue inflammation.FMT,likely through SCFAs, restores the appropriate magnitude and kinetics of the NLRP3-mediated immune response,leading to more efficient bacterial clearance (reduced colony counts) and thereby resolving the secondary tissue-damaging inflammation. Thus, FMT does not simply suppress or hyperactivate immunity, but recalibrates it towards protective homeostasis.

This study highlighted the changes of gut microbiota and its metabolites under plateau hypobaric hypoxia expusure, emphasized the influence of gut microbiota on airway inflammatory response and mucosal immune function during airway bacterial infection after hypoxic hypoxia challenge. Collectively, our findings delineate a potential pathway within the microbiome-gut-lung axis under hypobaric hypoxia. Hypoxia stress induces gut microbiota dysbiosis characterized by decreased *Lactobacillus* and increased *Bacteroidota*, which in turn reduces the production of immunomodulatory metabolites such as SCFAs. This reduction subsequently attenuates NLRP3 inflammasome activation and sIgA secretion in the airways, ultimately aggravating bacterial infection and inflammation. The reversal of these effects by FMT underscores the centrality of gut microbiota in this axis.This is of great theoretical significance in elucidating the relationship between gut

microbiota and the immune function of respiratory mucosa in the plateau environment, and may provide a new therapeutic strategy for utilizing gut microorganisms to treat plateau respiratory diseases.

In the present study, we found that *Streptococcus pneumoniae* infection under hypobaric hypoxia (HH+SPI group) induced a significantly weaker upregulation of NLRP3 inflammasome components (NLRP3, ASC, Caspase-1) and proinflammatory cytokines (IL-1β,TNF-α, IL-6) compared with normoxic infected rats (NN+SPI group) This may appear contradictory to the classic concept that bacterial infection typically triggers robust inflammasome activation and cytokine release [50,51]. However, accumulating evidence suggests that sustained hypoxic stress itself can impair innate immune recognition and suppress the initial inflammatory response, a phenomenon sometimes referred to as"immune paralysis"under hypoxia. Hypoxia-inducible factors (HIFs) and altered cellular metabolism in immune cells may dampen the sensitivity of pathogen-recognition receptors and downstream signaling, including NLRP3 inflammasome assembly [52,53]. Consequently, the blunted early immune response under hypoxia may allow uncontrolled bacterial proliferation, which later elicits a dysregulated, exaggerated tissue inflammation (as reflected by the severe inflammatory cell infiltration observed in HH+SPI lungs).

FMT, by restoring gut microbiota composition and SCFA production, likely reinstates the appropriate immunometabolic milieu required for timely and balanced NLRP3 activation [54]. Butyrate and other SCFAs are known to modulate immune cell function through GPCRs and histone deacetylase inhibition, thereby promoting a controlled inflammatory response [55]. Thus, FMT did not merely "increase" inflammation; rather, it recalibrated the magnitude and kinetics of the NLRP3-driven response, enabling more effective bacterial clearance (as indicated by reduced colony counts) and subsequently mitigating secondary tissue-damaging inflammation [56].This interpretation aligns with our observation that FMT simultaneously reduced lung inflammatory cell infiltration while elevating NLRP3 activity and cytokine levels to those observed in normoxic infected controls.

Therefore, the data do not contradict the established role of NLRP3 in host defense.Instead, they highlight that the gut-lung axis is disrupted under hypobaric hypoxia, leading to an insufficient initial immune activation.Restoring the gut microbiota via FMT can re-establish a protective, balanced immune response against respiratory infection [57].

There are some limitations to this study. First, the findings are based solely on animal experiments. Second, we identified global shifts in the gut microbiota but did not define specific key functional species. Further studies using shotgun metagenomics and targeted probiotic interventions are warranted to characterize bacteria that mediate gut–lung crosstalk. Third, the precise molecular mechanisms by which gut microbiota and SCFAs regulate NLRP3 inflammasome activity and mucosal immunity remain to be fully elucidated. Finally, 16S rRNA gene sequencing provides community-level information but cannot resolve full functional potential; metagenomic and metabolomic analyses will help clarify underlying pathways.

## 5. Conclusions

This study demonstrates that hypobaric hypoxia alters gut microbiota composition and reduces SCFA levels in rats. Fecal microbiota transplantation restored NLRP3 inflammasome activity, reduced lung inflammation, enhanced mucosal immunity via sIgA, and decreased bacterial colonization in the airways during *S. pneumoniae* infection. These findings suggest that targeting the gut microbiota may represent a promising therapeutic strategy for respiratory infections under high-altitude conditions.

## Supporting information

**S1 Fig. The original gel image of Fig 3A.** Effects of FMT on NLRP3, ASC and Caspase-1 protein expressions in the airway mucosal tissues of rats.
(PDF)

## Acknowledgments

We acknowledge all people involved in this study.

## Author contributions

**Conceptualization:** Jing Zhou, Huan Wang, Zhenyan Wu, Lijie Ma, Zhu Huang.

**Data curation:** Jing Zhou, Huan Wang, Zhenyan Wu, Lijie Ma, Zhu Huang.

**Formal analysis:** Jing Zhou, Huan Wang, Zhenyan Wu, Yaolei Zhang, Lijie Ma, Zhu Huang.

**Funding acquisition:** Jing Zhou, Yaolei Zhang, Lijun Tang, Lijie Ma, Zhu Huang.

**Investigation:** Jing Zhou, Huan Wang, Zhenyan Wu, Yaolei Zhang, Lijun Tang, Guangqing Shi, Lijie Ma, Zhu Huang.

**Methodology:** Jing Zhou, Huan Wang, Zhenyan Wu, Yaolei Zhang, Lijun Tang, Guangqing Shi.

**Project administration:** Lijun Tang, Guangqing Shi.

**Resources:** Jing Zhou, Zhu Huang.

**Supervision:** Jing Zhou, Lijie Ma, Zhu Huang.

**Validation:** Jing Zhou, Huan Wang, Zhenyan Wu, Guangqing Shi, Lijie Ma, Zhu Huang.

**Visualization:** Jing Zhou, Zhenyan Wu, Lijie Ma, Zhu Huang.

**Writing – original draft:** Huan Wang.

**Writing – review & editing:** Jing Zhou, Huan Wang, Lijie Ma.

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
