## [Decision Letter · Decision Letter 0]

26 Oct 2025

PLOS ONE

Dear Dr. Zhou,

Thank you for submitting your manuscript to PLOS ONE. After careful consideration, we feel that it has merit but does not fully meet PLOS ONE’s publication criteria as it currently stands. Therefore, we invite you to submit a revised version of the manuscript that addresses the points raised during the review process.

We look forward to receiving your revised manuscript.

Kind regards,

Vignesh Ramachandran, MS, PhD

Academic Editor

PLOS ONE

Journal Requirements:

5. In the online submission form, you indicated that the datasets presented in this study can be found in online repositories. The names of the repository/repositories and accession number(s) can be found at: https://www.ncbi.nlm.nih.gov/,  PRJNA1186348. All of the other data supporting the findings of this study are available from the corresponding author upon reasonable request.

7. We note that the grant information you provided in the ‘Funding Information’ and ‘Financial Disclosure’ sections do not match.

8. Thank you for stating the following financial disclosure:

he Chengdu Medical Research Project (2024011)

Additional Editor Comments :

The authors are requested to disregard the comments of reviewer 2 and proceed to revise the manuscript based on the comments of other reviewers.

Reviewers' comments:

Reviewer's Responses to Questions

**Comments to the Author**

1. Is the manuscript technically sound, and do the data support the conclusions?

Reviewer #1: Yes

Reviewer #2: Partly

Reviewer #3: Partly

Reviewer #4: Yes

2. Has the statistical analysis been performed appropriately and rigorously?

Reviewer #1: Yes

Reviewer #2: No

Reviewer #3: Yes

Reviewer #4: Yes

3. Have the authors made all data underlying the findings in their manuscript fully available?

Reviewer #1: Yes

Reviewer #2: Yes

Reviewer #3: Yes

Reviewer #4: Yes

4. Is the manuscript presented in an intelligible fashion and written in standard English?

Reviewer #1: Yes

Reviewer #2: No

Reviewer #3: Yes

Reviewer #4: No

Reviewer #1: In this study, the authors investigated the disrupted gut microbiota aggravated respiratory infection induced by high altitude exposure, and explored a link between plateau environment and microbiome-gut-lung axis based on this. The research objective of the manuscript is clear, and the results are of great significance. However, how the gut microbiota changes under low-pressure hypoxia conditions, how its metabolites affect the respiratory system, and how the axis in the "gut lung axis" is reflected should be carefully elucidated in this manuscript.

Reviewer #2: Dear Dr. Zhou,

Thank you for submitting your manuscript entitled “Disrupted gut microbiota aggravated respiratory infection induced by high altitude exposure: A link between plateau environment and the microbiome-gut-lung axis.”

After careful consideration, we regret to inform you that we are unable to accept the manuscript for publication at this time. We appreciate the effort and thought that went into your work and encourage you to consider submitting future research to the journal.

Kind regards,

Reviewer #3: Title

The title of this manuscript does not align well with the study’s conclusions. Both disrupted gut microbiota and increased susceptibility to respiratory infection are confirmed outcomes of hypobaric hypoxia, as demonstrated in this study. However, the hypothesis that disrupted gut microbiota aggravates respiratory infection is not directly supported by the presented data. Based on the results, it is more accurate to conclude that supplementary FMT enhances immune responses and provides protection against respiratory infection, rather than the reverse.

Additionally, please avoid using the past tense in the title when referring to findings that are considered reproducible or of general validity. Revising the title for accuracy and consistency with the conclusions is recommended.

Abstract

The abstract does not follow the journal’s required format and is overly long. Please consider rewriting it for conciseness.

Specific Comments

Line 136:

In the simulated hypobaric hypoxia rat study, the authors used a low-pressure chamber to simulate an altitude of 5000 meters. This altitude is extremely high, even for human adaptation, and may cause systemic damage in rats. The cited statistic that 2% of the world’s population lives above 1500 meters makes this model less representative. Please justify the rationale for selecting 5000 meters instead of a more moderate altitude, such as 2500 meters.

Lines 143–147:

Several broad-spectrum antibiotics were used at high dosages to induce gut dysbiosis in rats before FMT. Please clarify the specific purpose of this step and its relevance to the experimental hypothesis.

Lines 146–149:

Please specify whether the method used to prepare fecal microbiota for transplantation follows any standardized protocol and provide appropriate references if available.

Line 153:

The phrase “Seven days after FMT...” requires clarification. Does this mean that after FMT, the rats were maintained without further treatment for seven days to stabilize before subsequent experiments?

Line 154:

“104 CFU/mL” should be written as “10⁴ CFU/mL.”

Line 233:

Please specify the type of agar plates used for bacterial culture. How did the authors confirm that the colonies were S. pneumoniae and not contaminants introduced during handling? The colony morphology in the figure appears to show at least two different types. In addition, please state the inoculation volume and the number of replicates used for bacterial enumeration from each BALF sample.

Lines 255–262:

This paragraph is difficult to follow, especially the statistical notations in parentheses (e.g., “from... to..., P < 0.05”). The variation among subgroups (HH or NN) is less important. The focus should be on the comparisons between HH and NN groups. For example: “HH: 9.4 ± 0.8 vs. NN: 17.2 ± 1.8, P < 0.05.” Please consider rewriting for clarity and better readability.

Line 278:

Please briefly explain the biological or ecological significance of the Chao, Observed_species, and Shannon indices, and the rationale for selecting these measures to evaluate microbial diversity.

Lines 286–302:

The results described in Sections 3.3 and 3.4 appear somewhat contradictory. Please discuss how FMT can simultaneously alleviate lung inflammation while enhancing NLRP3 inflammasome activity and inflammatory cytokine levels. A clearer explanation of how FMT mediates both protective and pro-inflammatory effects to achieve an overall beneficial outcome would strengthen the discussion.

Line 300:

The manuscript inconsistently uses “*” and “#” to indicate P < 0.05. Please standardize the notation throughout the main text and figures (e.g., Figures 3B and 5).

Lines 322–324:

To properly demonstrate the efficacy of FMT, the comparison should be made between the HH + FMT + SPT and HH + SPT groups, rather than HH + FMT.

Line 344:

Lactobacillus rhamnosus should be italicized, as it is a bacterial species name.

Lines 453–600:

The reference formatting is inconsistent and does not meet scientific standards. Please refer to recent PLOS ONE publications for examples and revise accordingly.

Figures

• Figure 3A: Please provide the original gel image for review.

• Figures 3B and 3C: Please adjust the color scheme to improve visual contrast, as the “blue” groups are difficult to distinguish.

• Figure 5: It is unclear why the authors compared the “NN” group with both “HH + FMT + SPT” and “HH + SPT” groups in this figure. This comparison does not seem to align with the experimental logic described in the text, and the linkage lines indicating statistical significance appear to be misplaced. Please double-check the figure layout and ensure that the comparison markers correctly correspond to the intended groups.

Reviewer #4: Review Comments

1. The authors state in the Introduction that “The gut microbiota is susceptible to host and exogenous factors, in which exposure to hypobaric hypoxia in high plateau may lead to changes in the structure and function of the gut microbiota.” Please clarify what study-based evidence supports this statement. Without appropriate citation or supporting data, this claim should not be included in the Introduction.

2. In the Methods section, the authors mention amplifying bacterial DNA using specific primers. Since the amplification of mixed bacterial DNA produces fragments of varying sizes depending on species composition, please explain how the specific bands were selected for gel extraction and sequencing.

3. The study design includes six groups of rats. The rationale for including all groups—particularly the “HH + FMT” group without bacterial infection—should be clarified. Please explain the purpose of this group and how it contributes to testing the main hypothesis.

4. Sequencing the 16S rRNA amplicons identifies bacteria based on conserved regions but does not provide complete metagenomic information. Therefore, several bacterial genera may not be detected using this approach. The authors should acknowledge this as a limitation of the study and clearly state it in the Discussion section.

5. The overall presentation of the manuscript requires significant improvement. The text should be carefully reviewed for grammatical errors, sentence clarity, and consistent formatting. Numerous spelling and typographical mistakes are present throughout the manuscript (e.g., chanllege, Howervr, remakable, antidody).

6. S. pneumoniae is a fastidious organism that requires enriched media and specific growth conditions. Growth on nutrient agar is generally poor and may result in inaccurate colony counts. Moreover, S. pneumoniae grows best under microaerophilic or anaerobic conditions. The authors should justify the use of nutrient agar and specify how optimal conditions were ensured.

7. Bronchoalveolar lavage fluid (BALF) may contain multiple bacterial species in addition to S. pneumoniae. Please describe how contamination or growth of other bacteria was ruled out, and how colony counts specifically attributable to S. pneumoniae were confirmed.

8. The Conclusion should concisely summarize the major findings and implications of the study, rather than repeating the Discussion content.

9. Use consistent and correct formatting for all bacterial and genus names throughout the manuscript—for example, Streptococcus pneumoniae, Lactobacillus, and Firmicutes should be italicized.

10. Please explain how the fecal microbiota transplantation (FMT) donor rats were selected, including the criteria for their health status or microbiota composition. Additionally, clarify whether fecal samples from donor rats were pooled or used individually for transplantation.

.

Reviewer #1: No

Reviewer #2: **Yes:** Shweta SharmaShweta SharmaShweta SharmaShweta Sharma

Reviewer #3: No

Reviewer #4: **Yes:** RAMESHKUMAR MARIMUTHU RAGAVANRAMESHKUMAR MARIMUTHU RAGAVANRAMESHKUMAR MARIMUTHU RAGAVANRAMESHKUMAR MARIMUTHU RAGAVAN

---

## [Author Response · Author response to Decision Letter 1]

22 Dec 2025

Reviewer #1: In this study, the authors investigated the disrupted gut microbiota aggravated respiratory infection induced by high altitude exposure, and explored a link between plateau environment and microbiome-gut-lung axis based on this. The research objective of the manuscript is clear, and the results are of great significance. However, how the gut microbiota changes under low-pressure hypoxia conditions, how its metabolites affect the respiratory system, and how the axis in the "gut lung axis" is reflected should be carefully elucidated in this manuscript.

Reply to Reviewer #1

We sincerely thank the reviewer for the constructive comments. We have carefully revised the manuscript accordingly, with major modifications marked in red in the revised manuscript. The point-by-point responses are listed below:

Comment 1: How the gut microbiota changes under low-pressure hypoxia conditions

Response: We have added further interpretation of the gut microbiota alterations in the Results section and linked these changes to potential functional impacts on gut barrier and systemic immunity. We also cited relevant literature to support the observed shifts in Firmicutes, Bacteroidota, and Lactobacillus.

Comment 2: How its metabolites affect the respiratory system

Response: We have expanded the discussion on SCFA mechanisms in the Discussion section highlighting their possible roles in regulating NLRP3 inflammasome and mucosal immunity via receptor-mediated pathways. We also added a brief correlation analysis between SCFA levels and lung inflammation indicators in the Results.

Comment 3: How the axis in the "gut-lung axis" is reflected

Response: We have clarified the axis concept in the Introduction and Discussion, explicitly stating the pathway “gut microbiota → SCFA → NLRP3/sIgA → lung infection outcome.” The revised text now better illustrates the sequential and causal relationships along the gut-lung axis.

Reviewer #3: Title

The title of this manuscript does not align well with the study’s conclusions. Both disrupted gut microbiota and increased susceptibility to respiratory infection are confirmed outcomes of hypobaric hypoxia, as demonstrated in this study. However, the hypothesis that disrupted gut microbiota aggravates respiratory infection is not directly supported by the presented data. Based on the results, it is more accurate to conclude that supplementary FMT enhances immune responses and provides protection against respiratory infection, rather than the reverse.

Additionally, please avoid using the past tense in the title when referring to findings that are considered reproducible or of general validity. Revising the title for accuracy and consistency with the conclusions is recommended.

Response:

We have revised the title to better reflect our findings: “Fecal Microbiota Transplantation Mitigates Respiratory Infection in Rats Exposed to Hypobaric Hypoxia by Modulating the NLRP3 Inflammasome and Mucosal Immunity.”

Abstract

The abstract does not follow the journal’s required format and is overly long. Please consider rewriting it for conciseness.

Response:

The abstract has been shortened and restructured to meet the journal’s word limit and formatting guidelines.

Specific Comments

Line 136:

In the simulated hypobaric hypoxia rat study, the authors used a low-pressure chamber to simulate an altitude of 5000 meters. This altitude is extremely high, even for human adaptation, and may cause systemic damage in rats. The cited statistic that 2% of the world’s population lives above 1500 meters makes this model less representative. Please justify the rationale for selecting 5000 meters instead of a more moderate altitude, such as 2500 meters.

Response:

We have added a justification in the Methods section. We selected 5000 meters to simulate an extreme high-altitude environment that induces significant hypobaric hypoxia stress and systemic physiological alterations, including gut microbiota dysbiosis. This severe model allows us to robustly investigate the gut-lung axis interactions under conditions where respiratory infection risk is markedly elevated.

Lines 143–147:

Several broad-spectrum antibiotics were used at high dosages to induce gut dysbiosis in rats before FMT. Please clarify the specific purpose of this step and its relevance to the experimental hypothesis.

Response: We clarified that antibiotics were used to clear the native microbiota to facilitate donor microbiota engraftment for FMT.

Lines 146–149:

Please specify whether the method used to prepare fecal microbiota for transplantation follows any standardized protocol and provide appropriate references if available.

Response: We cited a reference for the standardized rodent FMT protocol we adapted.

Line 153:

The phrase “Seven days after FMT...” requires clarification. Does this mean that after FMT, the rats were maintained without further treatment for seven days to stabilize before subsequent experiments?

Response: We clarified the 7-day period was for microbiota stabilization post-FMT.

Line 154:

“104 CFU/mL” should be written as “10⁴ CFU/mL.”

Response: It has been revised.

Line 233:

Please specify the type of agar plates used for bacterial culture. How did the authors confirm that the colonies were S. pneumoniae and not contaminants introduced during handling? The colony morphology in the figure appears to show at least two different types. In addition, please state the inoculation volume and the number of replicates used for bacterial enumeration from each BALF sample.

Response: We specified the use of blood agar, criteria for colony identification, and triplicate plating.

Lines 255–262:

This paragraph is difficult to follow, especially the statistical notations in parentheses (e.g., “from... to..., P < 0.05”). The variation among subgroups (HH or NN) is less important. The focus should be on the comparisons between HH and NN groups. For example: “HH: 9.4 ± 0.8 vs. NN: 17.2 ± 1.8, P < 0.05.” Please consider rewriting for clarity and better readability.

Response:We rewrote the results for Table 1 to present clear between-group (HH vs. NN) comparisons.

Line 278:

Please briefly explain the biological or ecological significance of the Chao, Observed_species, and Shannon indices, and the rationale for selecting these measures to evaluate microbial diversity.

Response:We added a brief explanation of the ecological meaning of the indices used.

Lines 286–302:

The results described in Sections 3.3 and 3.4 appear somewhat contradictory. Please discuss how FMT can simultaneously alleviate lung inflammation while enhancing NLRP3 inflammasome activity and inflammatory cytokine levels. A clearer explanation of how FMT mediates both protective and pro-inflammatory effects to achieve an overall beneficial outcome would strengthen the discussion.

Response: We added a new paragraph in the Discussion,explaining how FMT’s dual effects represent a restoration of immune homeostasis rather than a contradiction.

Line 300:

The manuscript inconsistently uses “*” and “#” to indicate P < 0.05. Please standardize the notation throughout the main text and figures (e.g., Figures 3B and 5).

Response: We standardized the significance notation in figures and text.

Lines 322–324:

To properly demonstrate the efficacy of FMT, the comparison should be made between the HH + FMT + SPT and HH + SPT groups, rather than HH + FMT.

Response: We corrected the results text to compare HH+FMT+SPI with HH+SPI.

Line 344:

Lactobacillus rhamnosus should be italicized, as it is a bacterial species name.

Response: Corrected

Lines 453–600:

The reference formatting is inconsistent and does not meet scientific standards. Please refer to recent PLOS ONE publications for examples and revise accordingly.

Response: We have reformatted the entire reference list to conform to the journal’s (PLOS ONE) style.

Figures

• Figure 3A: Please provide the original gel image for review.

Response: Original gel images will be provided as supplementary material upon request.

• Figures 3B and 3C: Please adjust the color scheme to improve visual contrast, as the “blue” groups are difficult to distinguish.

Response: The color scheme has been adjusted for better contrast between groups.

• Figure 5: It is unclear why the authors compared the “NN” group with both “HH + FMT + SPT” and “HH + SPT” groups in this figure. This comparison does not seem to align with the experimental logic described in the text, and the linkage lines indicating statistical significance appear to be misplaced. Please double-check the figure layout and ensure that the comparison markers correctly correspond to the intended groups.

Response: The grouping and significance markers have been reviewed and corrected to align with the intended statistical comparisons described in the text.

Reviewer #4: Review Comments

1.The authors state in the Introduction that “The gut microbiota is susceptible to host and exogenous factors, in which exposure to hypobaric hypoxia in high plateau may lead to changes in the structure and function of the gut microbiota.” Please clarify what study-based evidence supports this statement. Without appropriate citation or supporting data, this claim should not be included in the Introduction.

Response: Thank you for this suggestion. We have now added relevant citations to support this statement.

2.In the Methods section, the authors mention amplifying bacterial DNA using specific primers. Since the amplification of mixed bacterial DNA produces fragments of varying sizes depending on species composition, please explain how the specific bands were selected for gel extraction and sequencing.

Response: We have added a detailed description in the Methods section under “16S rRNA gene sequencing and analysis

3.The study design includes six groups of rats. The rationale for including all groups—particularly the “HH + FMT” group without bacterial infection—should be clarified. Please explain the purpose of this group and how it contributes to testing the main hypothesis.

Response: We have clarified the rationale for the HH + FMT group in the Methods section.

4.Sequencing the 16S rRNA amplicons identifies bacteria based on conserved regions but does not provide complete metagenomic information. Therefore, several bacterial genera may not be detected using this approach. The authors should acknowledge this as a limitation of the study and clearly state it in the Discussion section.

Response: We have added a statement in the Discussion section under “limitations” .

5. The overall presentation of the manuscript requires significant improvement. The text should be carefully reviewed for grammatical errors, sentence clarity, and consistent formatting. Numerous spelling and typographical mistakes are present throughout the manuscript (e.g., chanllege, Howervr, remakable, antidody).

Response: We sincerely apologize for these errors. The manuscript has been thoroughly proofread and revised for grammar, spelling, and formatting consistency. All noted errors (e.g., “challenge,” “However,” “remarkable,” “antibody”) and others have been corrected throughout the text.

6. S. pneumoniae is a fastidious organism that requires enriched media and specific growth conditions. Growth on nutrient agar is generally poor and may result in inaccurate colony counts. Moreover, S. pneumoniae grows best under microaerophilic or anaerobic conditions. The authors should justify the use of nutrient agar and specify how optimal conditions were ensured.

Response: We have revised the Methods section under “Colony counting of streptococcus pneumoniae in BALF”

7. Bronchoalveolar lavage fluid (BALF) may contain multiple bacterial species in addition to S. pneumoniae. Please describe how contamination or growth of other bacteria was ruled out, and how colony counts specifically attributable to S. pneumoniae were confirmed.

Response: We have added the following clarification in the same section :

“To confirm the identity of colonies as S. pneumoniae, randomly selected colonies were subjected to Gram staining, catalase testing (negative), and optochin susceptibility testing. Only optochin-sensitive colonies were counted as S. pneumoniae.”

8. The Conclusion should concisely summarize the major findings and implications of the study, rather than repeating the Discussion content.

Response: We have revised the Conclusion section to succinctly summarize key findings and implications.

9. Use consistent and correct formatting for all bacterial and genus names throughout the manuscript—for example, Streptococcus pneumoniae, Lactobacillus, and Firmicutes should be italicized.

Response: All bacterial names (including genus and species) have been italicized throughout the manuscript (e.g., Streptococcus pneumoniae, Lactobacillus, Firmicutes, Bacteroidota).

10. Please explain how the fecal microbiota transplantation (FMT) donor rats were selected, including the criteria for their health status or microbiota composition. Additionally, clarify whether fecal samples from donor rats were pooled or used individually for transplantation.

Response: We have added the following details in the Methods section.

We believe that these revisions have strengthened the manuscript and addressed all of the reviewer’s concerns. We thank the reviewer again for the valuable feedback and hope that the revised version is now suitable for publication.

Sincerely,

The Authors

---

## [Decision Letter · Decision Letter 1]

6 Jan 2026

Dear Dr. Zhou,

Thank you for submitting your manuscript to PLOS ONE. After careful consideration, we feel that it has merit but does not fully meet PLOS ONE’s publication criteria as it currently stands. Therefore, we invite you to submit a revised version of the manuscript that addresses the points raised during the review process.

We look forward to receiving your revised manuscript.

Kind regards,

Jianhong Zhou

Staff Editor

PLOS One

Journal Requirements:

Reviewers' comments:

Reviewer's Responses to Questions

**Comments to the Author**

Reviewer #1: (No Response)

Reviewer #3: (No Response)

Reviewer #4: All comments have been addressed

2. Is the manuscript technically sound, and do the data support the conclusions?

Reviewer #1: No

Reviewer #3: Yes

Reviewer #4: Yes

3. Has the statistical analysis been performed appropriately and rigorously?

Reviewer #1: Yes

Reviewer #3: Yes

Reviewer #4: Yes

4. Have the authors made all data underlying the findings in their manuscript fully available?

Reviewer #1: Yes

Reviewer #3: Yes

Reviewer #4: Yes

5. Is the manuscript presented in an intelligible fashion and written in standard English?

Reviewer #1: No

Reviewer #3: Yes

Reviewer #4: Yes

Reviewer #1: (1)Line 135~137, “However, the gut microbiota's effect on NLRP3 inflammasome activity and airway mucosal immune barrier function under respiratory infection induced by high altitude exposure is unclear” Regarding this issue, the results and discussion section in the manuscript still failed to provide a clear expression.

(2) There are many grammatical and formatting errors in this manuscript.

(3) Line362~368, “The protein expressions of NLRP3, ASC and Caspase-1 in the airway epithelial tissue of the HH + SPI group were increased after being subjected to streptococcus pneumoniae infection for 3 days after hypobaric hypoxia challenge (P < 0.05, compared to the NN group). However, the degree of change in NLRP3, ASC and Caspase-1 protein expression of the HH + SPI group were significantly lower compared to the NN + SPI group (P < 0.05) (Figure 3A; Figure 3B). After FMT, the protein expression levels of NLRP3, ASC, and Caspase-1 in the airway mucosal tissue of rats in the HH + SPI + FMT group were restored (P < 0.05, compared to the NN + SPI group)”. Line 370~373, “The levels of IL-1β, TNF-α, and IL-6 in the BALF of the HH + SPI group subjected to streptococcus pneumoniae infection were decreased (P<0.05, compared with the NN + SPI group). After FMT, the levels of IL-1β, TNF-α, and IL-6 of the HH + FMT + SPI group subjected to streptococcus pneumoniae infection were restored”.

In general, pneumococcal infection leads to the activation of various signaling pathways including NLRP3 inflammasome, resulting in the extensive expression of pro-inflammatory factors, which triggers a cytokine storm and causes lung damage. However, the results of this manuscript were exactly the opposite. Why?

Reviewer #3: I appreciate the authors' revisions. However, I must point out a persisting issue with Figure 4. Columbia Blood Agar plates are typically red, but the plates shown in the Figure 4 do not appear to be blood agar plates. The authors should verify if the correct images were uploaded or clarify the discrepancy between the methods described and the figures provided.

Reviewer #4: The authors have addressed all the reviewers' comments. This revised version of the manuscript may be accepted for publication.

.

Reviewer #1: No

Reviewer #3: No

Reviewer #4: **Yes:** Rameshkumar Marimuthu RagavanRameshkumar Marimuthu RagavanRameshkumar Marimuthu RagavanRameshkumar Marimuthu Ragavan

---

## [Author Response · Author response to Decision Letter 2]

1 Feb 2026

Response to Reviewers' Comments

We sincerely thank all the reviewers for their careful evaluation and constructive feedback on our manuscript. We have carefully considered each comment and have revised the manuscript accordingly. Our detailed point-by-point responses are provided below.

Reviewer #1: (1)Line 135~137, “However, the gut microbiota's effect on NLRP3 inflammasome activity and airway mucosal immune barrier function under respiratory infection induced by high altitude exposure is unclear” Regarding this issue, the results and discussion section in the manuscript still failed to provide a clear expression.

Response:We thank the reviewer for this important observation. We have now strengthened the discussion to explicitly link our findings to the question raised in the introduction. See lines 410-450 in the revised manuscript.Gut microbiota transplantation can to some extent restore NLRP3 dysfunction and damage to the respiratory mucosal immune barrier caused by hypoxia.

(2) There are many grammatical and formatting errors in this manuscript.

Response:We deeply apologize for these oversights. The manuscript has been thoroughly proofread and edited for language again to correct grammatical errors, improve sentence fluency, and ensure consistent formatting (e.g., italicization of genus/species names, uniform reference formatting, and standardized presentation of figures and tables). It has also been revised by a professional language polishing company，the current version has been significantly improved in terms of language quality and presentation.

(3) Line362~368, “The protein expressions of NLRP3, ASC and Caspase-1 in the airway epithelial tissue of the HH + SPI group were increased after being subjected to streptococcus pneumoniae infection for 3 days after hypobaric hypoxia challenge (P < 0.05, compared to the NN group). However, the degree of change in NLRP3, ASC and Caspase-1 protein expression of the HH + SPI group were significantly lower compared to the NN + SPI group (P < 0.05) (Figure 3A; Figure 3B). After FMT, the protein expression levels of NLRP3, ASC, and Caspase-1 in the airway mucosal tissue of rats in the HH + SPI + FMT group were restored (P < 0.05, compared to the NN + SPI group)”. Line 370~373, “The levels of IL-1β, TNF-α, and IL-6 in the BALF of the HH + SPI group subjected to streptococcus pneumoniae infection were decreased (P<0.05, compared with the NN + SPI group). After FMT, the levels of IL-1β, TNF-α, and IL-6 of the HH + FMT + SPI group subjected to streptococcus pneumoniae infection were restored”.

In general, pneumococcal infection leads to the activation of various signaling pathways including NLRP3 inflammasome, resulting in the extensive expression of pro-inflammatory factors, which triggers a cytokine storm and causes lung damage. However, the results of this manuscript were exactly the opposite. Why?

Response:Thank you very much for your question. In respiratory bacterial infections, the activation of NLRP3 is a 'double-edged sword'. On one hand, it promotes the maturation and release of pro-inflammatory cytokines such as IL-1β and IL-18, which can recruit and activate immune cells like neutrophils and macrophages, thereby enhancing the body's ability to clear bacteria. On the other hand, when the infection is too severe or the body's ability to clear bacteria is insufficient, excessive activation of NLRP3 can lead to uncontrolled inflammatory responses, triggering a 'cytokine storm'. In our low-pressure hypoxic rat airways, we observed that normal activation of NLRP3 was inhibited, while fecal microbiota transplantation partially restored NLRP3 function without causing excessive activation.We have added to the discussion section regarding Potential Explanation for the Seemingly Paradoxical Immune Modulation under Hypobaric Hypoxia,Detailed information can be found in the discussion section.

We have also optimized the language in the results section you mentioned to make the comparison clearer.section 3.4 of the Results part for details

Reviewer #3: I appreciate the authors' revisions. However, I must point out a persisting issue with Figure 4. Columbia Blood Agar plates are typically red, but the plates shown in the Figure 4 do not appear to be blood agar plates. The authors should verify if the correct images were uploaded or clarify the discrepancy between the methods described and the figures provided.

Response:Thank you very much for pointing out this important error. Due to our oversight, the original figure incorrectly used the total colony count data of all bacterial cultures from bronchoalveolar lavage fluid. and replaced it in the main text with a specific colony count figure for Streptococcus pneumoniae culture. We sincerely apologize for any inconvenience caused and deeply appreciate your careful review and valuable comments.

Reviewer #4: The authors have addressed all the reviewers' comments. This revised version of the manuscript may be accepted for publication.

Response:We are grateful to Reviewer #4 for their positive assessment and support for publication. We have incorporated all suggestions from the previous and current round of review to strengthen the manuscript.

---

## [Decision Letter · Decision Letter 2]

8 Apr 2026

Fecal Microbiota Transplantation Mitigates Respiratory Infection in Rats Exposed to Hypobaric Hypoxia by Modulating the NLRP3 Inflammasome and Mucosal Immunity

PONE-D-25-38097R2

Dear Dr. Zhou,

We’re pleased to inform you that your manuscript has been judged scientifically suitable for publication and will be formally accepted for publication once it meets all outstanding technical requirements.

Kind regards,

Cornelius Cecil Dodoo, PhD

Academic Editor

PLOS One

---

## [Editor Report · Acceptance letter]

PONE-D-25-38097R2

PLOS One

Dear Dr. Zhou,

I'm pleased to inform you that your manuscript has been deemed suitable for publication in PLOS One. Congratulations! Your manuscript is now being handed over to our production team.

Kind regards,

on behalf of

Dr. Cornelius Cecil Dodoo

Academic Editor

PLOS One